# Projection of COVID-19 Positive Cases Considering Hybrid Immunity: Case Study in Tokyo

**DOI:** 10.3390/vaccines11030633

**Published:** 2023-03-13

**Authors:** Sachiko Kodera, Akito Takada, Essam A. Rashed, Akimasa Hirata

**Affiliations:** 1Department of Electrical and Mechanical Engineering, Nagoya Institute of Technology, Nagoya 466-8555, Japan; 2Center of Biomedical Physics and Information Technology, Nagoya Institute of Technology, Nagoya 466-8555, Japan; 3Graduate School of Information Science, University of Hyogo, Kobe 650-0047, Japan

**Keywords:** hybrid immunity, vaccination effectiveness, herd immunity, COVID-19, forecasting, deep learning

## Abstract

Since the emergence of COVID-19, the forecasting of new daily positive cases and deaths has been one of the essential elements in policy setting and medical resource management worldwide. An essential factor in forecasting is the modeling of susceptible populations and vaccination effectiveness (VE) at the population level. Owing to the widespread viral transmission and wide vaccination campaign coverage, it becomes challenging to model the VE in an efficient and realistic manner, while also including hybrid immunity which is acquired through full vaccination combined with infection. Here, the VE model of hybrid immunity was developed based on an in vitro study and publicly available data. Computational replication of daily positive cases demonstrates a high consistency between the replicated and observed values when considering the effect of hybrid immunity. The estimated positive cases were relatively larger than the observed value without considering hybrid immunity. Replication of the daily positive cases and its comparison would provide useful information of immunity at the population level and thus serve as useful guidance for nationwide policy setting and vaccination strategies.

## 1. Introduction

Severe acute respiratory syndrome coronavirus 2 (SARS-CoV-2) virus, which emerged in late 2019, has since spread globally. Unlike upsurges in 2020 and 2021, the number of death cases per population has become small within the spread of the Omicron variant and its sublineage [1]. In most European and American countries, almost no movement restriction has been implemented since early 2022, whereas in various East Asian countries, some preventive measures or precautions were still implemented into late 2022 [2,3,4].

The percentage of people infected with SARS-CoV-2 in European countries is significant. This may be partly attributable to reinfections [5,6,7]. In such countries, the number of infected people during a new surge becomes smaller than that in earlier waves [1]. Instead, in East Asian countries, the number of new cases grew until the Omicron variant and its subvariants (until late 2022). The abovementioned difference in policy or regulation setting would result in immunity at the population level. Infection, including reinfection, may result in immunity acquisition in addition to the vaccination [8].

Several studies have reported that hybrid immunity is the most robust immunity against COVID-19 infection [9,10,11,12]. Hybrid immunity is defined as the immunity of vaccinated individuals, with two to three doses, alongside a primary SARS-CoV-2 infection. Carazo et al. [13] showed that hybrid immunity from BA.2 infection plus two to three doses of the vaccine similarly improved the estimated individual vaccination effectiveness (VE) to 96% longer than 5 months. However, the study was conducted with a limited scope for a group of healthcare workers. In [14], the booster shots combined with infection were reported to cause higher protection from infection with a milder waning effect than the booster shots alone. The effectiveness of infection prevention in individuals is attenuated with time and depends on the timing of primary infection. However, few studies have modeled hybrid immunity with time [15], resulting in difficulty in its application to vaccination strategies. The number of reported daily positive cases (DPC) has been inhibited since August 2022 in Japan, where the Omicron BA.5 variant is dominant. Unlike earlier waves of COVID-19 decay and respread (11% of the total population before the sixth wave of the pandemic), the number of infected people reached 21 million (corresponding to 23% of the entire population in Japan) [16].

For the estimation of VE at the population level, the modeling of the individual hybrid immunity is one of the key factors in forecasting the number of newly positive cases [17] where a high percentage of people are both vaccinated and primarily infected. This estimation is also useful for developing vaccine strategies [18]. Questions that remain are (1) to what degree have people obtained hybrid immunity, and (2) how does it affect the DPC in the real world. One approach to estimate the VE is to replicate the DPC using an estimated VE based on in vitro antibody measurements.

This study aimed to develop a mathematical VE model at the population level that considers the effect of hybrid immunity. Its effectiveness is confirmed from the replication of the DPC using data acquired from Tokyo between June 2021 and October 2022, which coincided with the fifth wave (from June to September 2021), the sixth wave (from January to May 2022), and the seventh wave (from July to September 2022) in Japan.

## 2. Materials and Methods

### 2.1. Materials

The Vaccination Record System of the Digital Agency [19] provided the number of newly vaccinated individuals per day, as shown in Figure 1a. The open dataset was divided into categories of dose number, gender, and age (binary of younger or older than 65 years old).

The Tokyo Metropolitan Agent provided the DPCs, which were divided into fully vaccinated and nonvaccinated individuals [20], until 27 September 2022. After that day, the Ministry of Health, Labour, and Welfare [16] provided the DPCs.

In the Japanese Cabinet Secretariat COVID-19 AI & Simulation Project [21], three metrics, which related to human behavior, were considered: (1) mobility at the transit stations; (2) nighttime population who stayed in the downtown area, including restaurants and bars [22]; and (3) Twitter keywords (social gathering for drinking and BBQ). The number of tweets with the Twitter keywords (social gathering for drinking, karaoke, and BBQ) was considered as a metric to correlate with social behavior, as demonstrated in our previous study [23]. Here, (1) and (3) were considered because (2) was correlated to the remaining two factors. Mobility data were obtained from Google Mobility [24]. Mobility is defined as the percent difference in population volume at transit stations compared to a baseline. The baseline represents the median value for that day of the week from the 5-week period of 3 January to 6 February 2020 (before COVID-19 pandemic). Twitter data were obtained from NTT Data, Inc.; processed by the Toyoda Lab., University of Tokyo; and shared through the Cabinet Secretariat COVID-19 AI & Simulation Project. Tweeted keywords completed during the day, the previous day, or those planned for the next day were extracted when determining the number of tweeted keywords. Time series data for mobility and the number of tweets is summarized in Figure 1b.

To estimate the hybrid immunity in the real world, the number of individuals with asymptomatic infection also needed to be taken into consideration. The Bureau of Social Welfare and Public Health of the Tokyo Metropolitan has conducted free reverse transcription polymerase chain reaction (RT-PCR) testing on people without subjective symptoms, mainly in downtown areas, restaurants, and train stations [25]. The rate of asymptomatically infected individuals in relation to the reported positive cases was estimated by comparing both the positive rate of PCR testing in people without subjective symptoms and the reported positive rate against the total population [26], as shown in Figure 1c. The occupancy rate of SARS-CoV-2 by variants is also shown in Figure 1d. The ratio of asymptomatic infection was an estimated 3.9 times higher (95% CI: 3.0–7.0 times) than reported DPC between 1 September 2020 and 31 March 2021, from the first to the fourth wave [27]. The ratio shown in Figure 1c is comparable in the sixth wave, whereas the ratio was higher in the seventh wave. One potential reason for this difference is the pathogenicity and transmissibility of viral variants.

### 2.2. Vaccination Effectiveness (VE)

#### 2.2.1. Individual Vaccination Effectiveness

In this study, VE was defined as VE = 1—relative risk in the real world without controlling the conditions [28]. The relative risk is defined as the ratio of vaccinated and unvaccinated population among infected people. A point of emphasis here is that, unlike the vaccination efficacy, which is derived under controlled (rather ideal) conditions, the VE is affected by the behavior at the population level. The individual VE of each dose was represented mathematically as in our previous study [23,29] as follows:(1)$$e\left(i\right)=\left\{\begin{array}{c} a_{t}\cdot i/K\left(i\leq K\right)\\ a_{t}-s\left(i-K\right)\left(i>K\right) \end{array}\right.,$$
where parameters of *a_t_* and *s* were adjusted to reach a peak *K* days after the inoculation of *t*-th dose (*K* = 14 for the second dose and *K* = 7 for the third and subsequent doses) then decrease linearly. The parameters were the same as in [23]. For this model, the estimated individual VE, including its waning effect, is in good agreement with cohort studies in Tokyo and its suburb area [30,31].

Qu et al. [14] reported that the waning effect in the neutralizing antibody, which would be related to the vaccination efficacy, of an individual with a prior infection was slower than the waning effect in an individual with no prior infection. Antibody effectiveness and durability are assumed to increase because of hybrid immunity (third dose plus infection) based on this study, as indicated in Figure 2 for individual immunity (*e* in Equation (1)). The relationship between the antibody and vaccination efficacy is approximately derived as in a previous study [32].

The waning immunity in fully vaccinated individuals with primary infection, fully vaccinated individuals with no prior infection, and nonvaccinated individuals with primary infection was assumed to be −0.11%, −0.15%, and −0.15% per day, respectively [14]. Natural immunity due to primary infection was assumed to be equivalent to the first vaccination for each variant [33]. Other studies suggested that natural immunity may persist longer and be higher than assumed here [15,34]. In contrast, the immunity acquired by the asymptomatic infection is smaller than that of symptomatic infection and the values are not always consistent [35,36]. Considering this limitation regarding the asymptomatic infection, we set the initial individual VE acquired by infection as smaller than that of the full vaccination. In addition, immunity enhancement due to hybrid immunity is empirically assumed to be 60% or 80% of the ideal hybrid immunity from booster shot immunity. All parameters used in this study are listed in Table 1. In particular, the number of tweets related to social gathering is highly correlated with real-world VE; some drops in VE were observed during active social gathering [23].

#### 2.2.2. Population Vaccination Effectiveness

Population VE is an essential factor for estimating viral transmission [37,38,39,40,41]. The herd immunity threshold for SARS-CoV-2 was 50–83%, which is approximately derived from the basic reproduction number [37]. For the available data, the population VE for symptomatic infection in the convolution integral of the individual VE and the number of newly vaccinated people was derived as follows [29]:(2)$$E_{v}\left(d\right)=\frac{1}{P}\sum_{i=0}^{d}\sum_{t=1}^{T}\sum_{s}N_{t,}\left(d-i\right)\cdot e_{t}\left(i\right),$$
where *d* is the day index and *P* is the population of Tokyo (13,843,329 people). *N_t_* denotes the daily number of people who are inoculated with the *t*-th dose, and *e_t_* denotes the individual VE of the *t*-th dose.

Considering the hybrid immunity, a proposal was made to extend Equation (2) as follows:(3)$$E_{hy}\left(d\right)=\frac{1}{P}\sum_{i=0}^{d}\sum_{s}v_{s}\left\{N_{n,pi,}\left(d-i\right)\cdot e_{n,pi}\left(i\right)+\sum_{t=1}^{T}\left(N_{t,pi,}\left(d-i\right)\cdot e_{t,pi}\left(i\right)+N_{t,ni,}\left(d-i\right)\cdot e_{t,ni}\left(i\right)\right)\right\},$$

The first and second terms of the right-hand side in Equation (2) correspond to the natural immunity of nonvaccinated people with a primary infection and the summation of the hybrid immunity for vaccinated people with a primary infection and the immunity for vaccinated people with no previous infection. The parameter *e* denotes the individual immunity shown in Figure 2. *N_n,pi_* denotes the number of nonvaccinated people with primary infection. *N_t,pi_* denotes the number of the *t*-th vaccinated individuals with primary infection (*t* = 1–4). *N_t,ni_* is the number of the *t*-th vaccinated individuals with no previous infection. The *v_s_* denotes the occupancy rate of SARS-CoV-2 by variants shown in Figure 1c, accounting for difference in waning immunity due to changes in the predominant variants. The waning effect was overwritten when people took the booster dose by adjusting the number of people vaccinated in the past. Owing to the lack of data in Tokyo, *N_n,pi_*, *N_t,pi_*, and *N_t,ni_* were approximated by the rate of infected individuals using the vaccination history of Japan reported by the Adversary Board of the Ministry of Health, Labour, and Welfare [42]. The vaccination rate in each dose and estimated population VE is shown in Figure 3.

The observed data in Figure 3 is empirically derived from the ratio of vaccinated and unvaccinated populations among infected people. The observed data in Japan and Tokyo were estimated by using the datasets which are provided by the website of the Ministry of Health, Labor, and Welfare of Japan [43] and the press release by the Tokyo Metropolitan Government [44], respectively. The datasets include the number of unvaccinated individuals, fully vaccinated individuals, and those vaccinated with a booster dose, for the number of infected individuals in each category from April 2022 to September 2022. The dataset in [43] is the weekly information for all of Japan, and the dataset in [44] is the daily information in Tokyo, which is available only for Tokyo in Japan. The difference between the reported values and the mathematically estimated values becomes larger, especially in the Tokyo data after the number of infected people increases (after 30 June 2022).

### 2.3. Forecasting of Daily Positive Cases with Deep Neural Network

Many prediction models, such as SIR/SEIR and deep neural networks have been proposed to predict new positive cases during the epidemics. Although these predictions yield useful information, the prediction accuracy may depend on the quality of the data used to calibrate it. In addition, when contact is suppressed under the state of emergency, the modeling or parameter extraction is not as straightforward as is reported by multi-agent simulation [45,46].

A deep neural network approach is often employed for forecasting the DPC and death cases. The major contribution of deep neural networks is that they allow researchers to understand the correlation between various factors and the incidence of infection and death cases. The network architecture we proposed in [47] is a multipath neural network with long short-term memory modules (LSTM) (primarily used for time series forecasting) and fully connected layers (primarily used for learning correlation features) in two major phases as shown in Figure 2 of [17]. The design of the deep learning model is based on ablation study in [47].

The training process is performed as shown in Figure 4 of [48]. The input values detailed above for a certain period (14 days) are used along with the target output value in the successive period (14 days) to train the model. The training is conducted through minimizing cross-entropy loss function using the Adam algorithm for 1000 epochs and an automatically estimated training rate. The process is implemented using a workstation with 4 Intel ^®^ Xeon CPUs running at 3.6 GHz with 128 GB memory and a set of 3 NVIDIA GeForce 1080 GPUs. A single training session requires approximately 5 min.

Deep learning models are commonly presented as a black box, where the contribution of different factors is unknow. To avoid this weakness as much as possible, we have derived the population-level immunity mathematically as mentioned in the above subsections. In [23], the optimal combination of input values were investigated among mobility, meteorological data, parameters related to social behavior, the day labels, the population VE, and variant infectivity, whose selection was based on the correlation of the morbidity with several potential factors [49,50]. The set of input data are arranged in a 2D array where the x-axis represents the time-series data, and the y-axis is the value of different input variables. The target output is set to be the DPC associated with each date. The following were selected as input values from our analysis in terms of the mean absolute percentage error (MAPE) by comparing the reported and estimated DPC: mobility at the transit stations; the number of tweets with risk keywords in Twitter; population VE, which was obtained from the individual VE and vaccination rates [23]. Other factors which may potentially influence viral transmission include meteorological data, which was discussed extensively in the early stages of the pandemic [51,52,53], and the correlation between viral transmission and human behavior, which was reported in the later stages of the pandemic during the vaccination campaigns and the emergence of viral variants [54,55]. According to [23], their contributions are marginal in mid- and long-term projections, compared to the other factors. The association between meteorological factors and viral infection may be closely related to social behavior [52]. This may be implicitly included in behavior as infections occur inside buildings in metropolitan areas. Thus, the meteorological factor was not considered here.

Using this deep learning model enables an accurate forecasting of DPCs in three major urban areas of Japan (Tokyo, Osaka, and Aichi), which was provided in the COVID-19 AI & Simulation Project powered by the Cabinet Secretariat in Japan [21]. The validation of our model can be found in [21]; also see [17,23,29]. In this study, to evaluate the estimation accuracy of DPC, the mean absolute error (MAE) and MAPE were used.

## 3. Results

The numerical examples in 15 timeframes (the 1st week of every two months) are presented in Figure 4a to demonstrate the robustness of our computational approach in replicating the DPC. This includes the period when hybrid immunity was not expected (from the fifth to the seventh waves), i.e., the number of infected populations was rather small. Mobility and the number of tweets in the future are assumed to be known, and the reported DPCs were used for validation. Even in the periods with varying predominant variants and VE, a high consistency was observed between the estimated and observed values in almost all timeframes, except for one timeframe from 1 June 2022, even in the period of difference in dominant variant and vaccine effectiveness. The timeframe from 1 June 2022, corresponds to a less predictable period when the predominant variant alternated between Omicron BA.2 and BA.5 (see Figure 1c). Table 2 shows the MAEs and MAPEs across 15 timeframes. Except for the timeframe 3–6 when the average number of DPCs was less than 500, the average MAPE was 26.5% in the 1-month forecast, which increased to 48.8% in the 2-month forecast.

Our forecasting model in the above figure was focused during the seventh wave (30 June to 31 September 2022) with and without consideration of hybrid immunity. As is shown in Figure 4b, the immunity at the population level is influenced by the modeling of hybrid immunity. From Figure 4b, the estimated DPC with population VE considering hybrid immunity is in better alignment with reported values than those values without hybrid immunity. The MAPEs with hybrid immunity (high), (low), and without hybrid immunity were 14.0%, 108.3% and 242.0%, respectively, from 15 July 2022 to 15 October 2022.

## 4. Discussion and Conclusions

It is crucial to forecast the DPC to set policies and medical resource management. Unlike earlier waves, when the DPC number was smaller than the entire population, the population immunity became complex owing to infection and vaccination, in addition to new variant emergence. Modeling hybrid immunity is one of the elements that will help to improve the DPC forecasting. To evaluate the impact of hybrid immunity at the population level, we estimated the DPC while taking into account the hybrid immunity with machine learning.

The feature of our model is that the number of input parameters is limited to four. Specifically, first, the population VE was estimated as pre-processing and validated by comparing daily reports of Tokyo (Figure 3b). The hybrid immunity was then proposed to be modeled for the first time based on previous in vitro studies [13,14]. The remaining factors considered in the model are viral infectivity, mobility at the transit stations, and the number of tweets with risk keywords on Twitter, which would be associated with human contact. Then, the effect of latency (mainly for incubation time) is considered as LSTM. Thus, our model would be close to nonlinear multivariate analysis considering the time shift in terms of virtually infinite fitting parameters in machine learning.

The estimated population VE was in good agreement with the reported value during the quasi-state of emergency (until March 2022). After the state was lifted, some drops in the reported population VE were seen compared to the estimated VE with the increasing in social gatherings [23]. A non-negligible population was infected in the seventh wave for the first time in Tokyo, but not in the suburb area of Japan (over Japan). The population VE in the real world was affected by social behavior [56]. In Tokyo, social behavior was curtailed due to the quasi-state of emergency until 21 March 2022.

The population VE in Tokyo was derived based on the vaccination history and infection rates. The comparison of replicated and observed DPCs was conducted to verify our modeling for the period of the seventh wave, in which the number of positive cases cannot be ignored; that is, the effect of hybrid immunity became significant. As shown in Figure 4b, the forecasting with hybrid immunity at a high level is a better explanation of the seventh wave. Here, we assumed that the vaccination effectiveness for infection prevention was 80% of the ideal value, which is derived from in vitro studies. This would hypothesize that the people with asymptomatic infection would acquire immunity with 60% effectiveness or approximately 60% of the population with asymptomatic infections. The threshold of herd immunity may be derived by comparing between the observed DPCs and estimated population VE; approximately 40%. The results suggested that the threshold of herd immunity can be achieved via a hybrid immunity [57], although further investigation is needed. Without considering the hybrid immunity, the MAE and MAPE were 15,389 and 242%, respectively, from 15 July to 15 October 2022, which were larger than those in earlier waves (see Figure 4b). The limitations of this study are as follows:The DPC does not include all infected populations, because it is only the “reported” cases. The population with asymptomatic infections, as well as the limited capacity of the tests, must be taken into consideration. The former was considered, approximately, as is shown in Figure 1b, while it cannot be considered for the latter;For long-term forecasting (its replication), the weather and holiday behavior were not considered. Meteorological data is shown to be associated with the morbidity and mortality rates [58,59]. The population VE is affected by social behavior [56], which may partly explain the VE difference between different countries [60,61,62]. This long-term modeling may provide general applicability of modeling, whereas some information may be potentially ignored. Thus, further fine-tuning of the hybrid immunity would not improve the accuracy of the modeling due to other uncertainty factors;The reinfection immunity, as well as pre-existing immunity levels to OC43, HKU1, and coronaviruses, were not considered. This may also be related to immunological imprinting of the antibody response [63]. Unlike most other countries, this imprinting effect was almost negligible until the seventh wave as the percentage of the estimated infected population was 11% of the total population. If this is existent, the number of estimated DPC would be higher. However, this is well within the range how the hybrid immunity can provide the VE. In this study, it was empirically assumed as 60% or 80% of the ideal hybrid immunity from booster shot immunity including the uncertainty of the immunity of asymptomatic infection (see Section 2.2.1);For simplicity, the natural immunity obtained due to infection in unvaccinated people was assumed to wane at the same slope as with vaccination; although, it is possible that the immune effect could persist for a longer period [34]. However, in Japan, where vaccination rates have exceeded 80%, the impact of immunity in unvaccinated people with a primary infection on predictions of DPCs is small especially before the seventh wave. In addition, the percentage of the infected population compared to the unvaccinated population was not large until the sixth wave. As mentioned in Section 2.2.1, people with asymptomatic infection are also considered, and thus VE was set lower than fully vaccinated people. However, this simplification may not result in significant error due to the fact that the number of infected people prior to the period considered here was small as and a high percentage of the population of Japan was vaccinated.

In conclusion, immunity modeling becomes more important in future forecasting and replication than in the earlier waves. The results of this study suggested that DPC monitoring in the real world would provide insight into the expectations of hybrid immunity and its durability based on limited data. The model proposed here would be helpful for policy setting and vaccination strategies.

## Figures and Tables

**Figure 1 vaccines-11-00633-f001:**
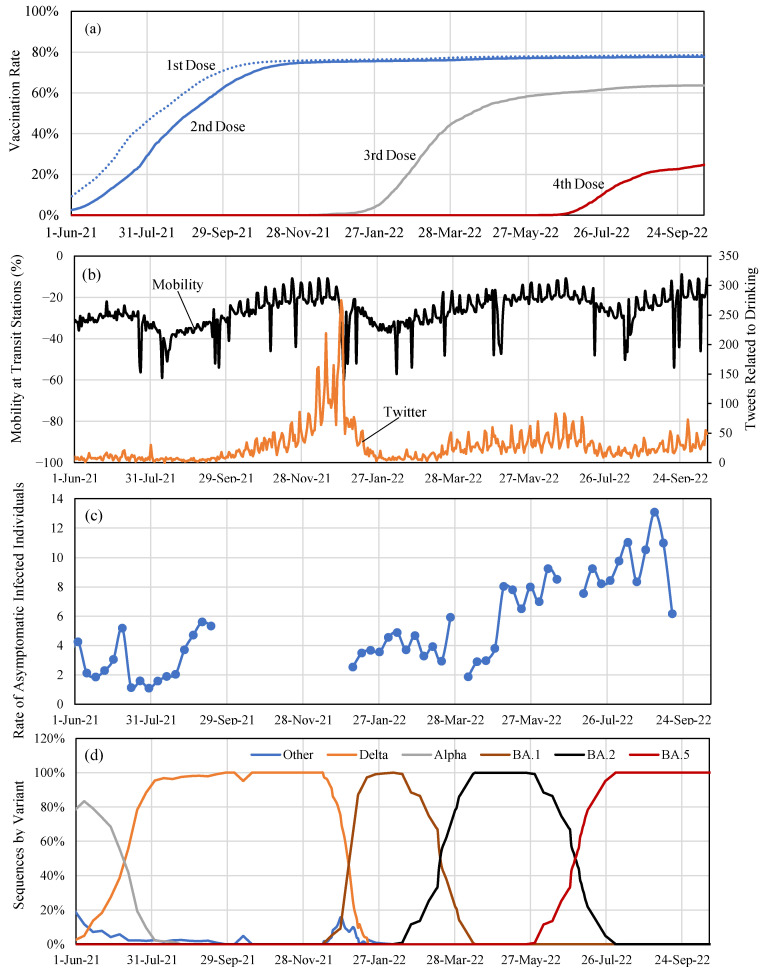
Time series of the (**a**) vaccination rates, and (**b**) the mobility at transit stations and the number of tweets related to drinking on Twitter. (**c**) Rate of asymptomatically infected individuals to reported positive cases. Missing period data are from 28 March to 3 April 2022, and 27 June to 3 July 2022. Periods with <500 average daily positive cases are not shown owing to a high level of uncertainty. (**d**) Time sequences by SARS-CoV-2 variants.

**Figure 2 vaccines-11-00633-f002:**
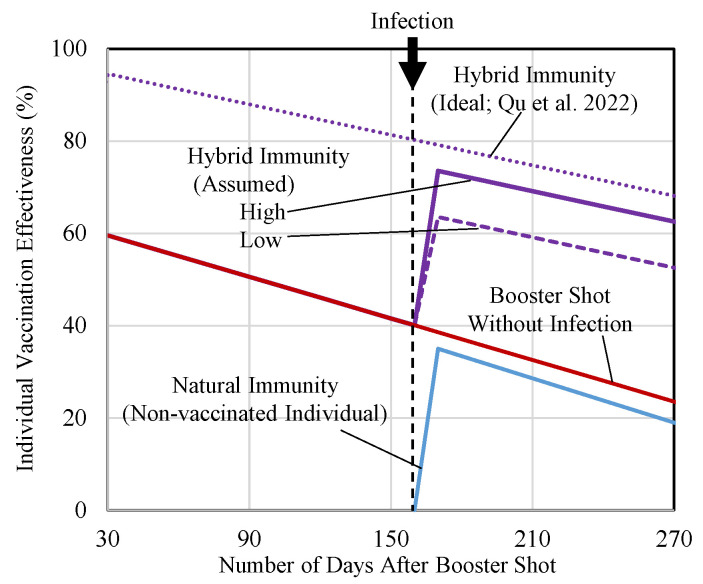
Waning immunity of booster with and without previous SARS-CoV-2 infection [14].

**Figure 3 vaccines-11-00633-f003:**
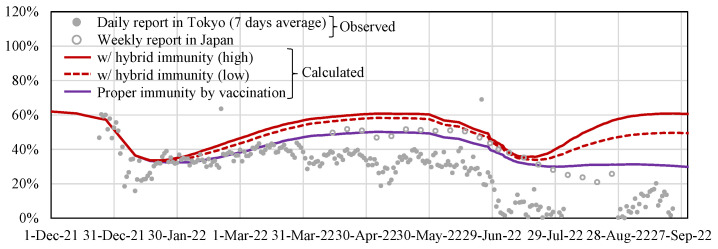
Time series of estimated population vaccination effectiveness (VE) in Tokyo from 1 July to 30 September 2022. Empirical estimation from reported DPC and their vaccination status is also plotted. This report ended 26 September 2022 in Tokyo and 28 August 2022 in Japan.

**Figure 4 vaccines-11-00633-f004:**
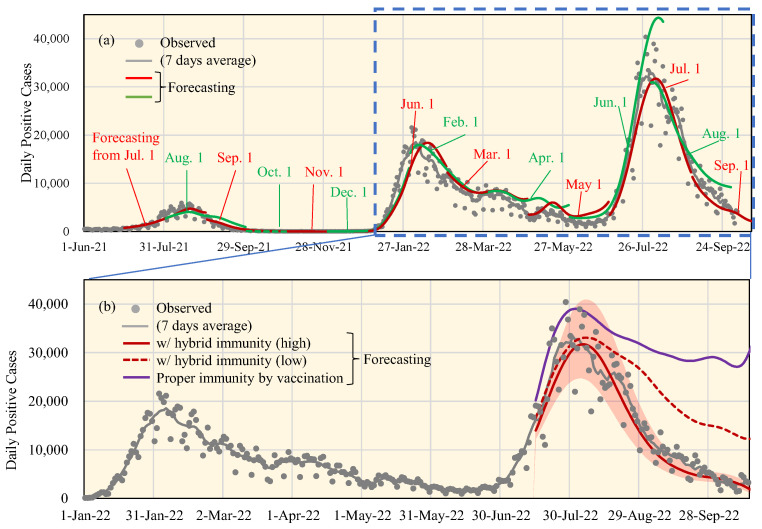
(**a**) Performance of the forecasting system for daily positive cases. In evaluating the performance, Twitter and mobility data were assumed to be known from the reported value. A 2-month forecast was made every month (on the first day of the month). The vaccination effectiveness with hybrid immunity (High), shown in (**b**), was considered. Observed cases and seven-day average values are shown in grey. Red and green curves are used to represent the forecasting to clarify the overlap. (**b**) Time series of forecasting of the daily positive cases in Tokyo from 1 July to 30 September 2022, considering the vaccination and hybrid immunity at the population level. The red colored area indicates the 95% confidence interval of the prediction considering hybrid immunity (High).

**Table 1 vaccines-11-00633-t001:** Parameters for the individual VE used in Equation (1).

		Vaccine				Hybrid Immunity		Natural Immunity
Parameters		First Shot	Second Shot	Third Shot	Forth Shot	High	Low	
*a*	Delta [29]	75	96	–	–	73	62	35
	Omicron (BA.1) [23]	35	63	85	–			
	Omicron (BA.2) [23]	–	61	86	–			
	Omicron (BA.5) [23]	–	37	63	69			
*s*		0.15	0.15	0.15	0.15	0.11	0.11	0.15

**Table 2 vaccines-11-00633-t002:** MAEs and MAPEs across 15 timeframes of forecasting daily positive cases shown in Figure 4a before the seventh wave.

			One Month		Two Month	
Timeframe	Start Date	End Date	MAE	MAPE	MAE	MAPE
1	1 July 2021	31 August 2021	311.8	25.7%	404.2	19.2%
2	1 August 2021	30 September 2021	654.9	15.3%	922.8	105.4%
3	1 September 2021	31 October 2021	349.4	58.3%	204.7	84.8%
4	1 October 2021	30 November 2021	36.2	52.6%	24.6	61.9%
5	1 November 2021	31 December 2021	7.5	40.1%	14.6	50.5%
6	1 December 2021	31 January 2022	7.7	22.7%	916.5	24.0%
7	1 January 2022	28 February 2022	2088.2	33.1%	2573.6	27.4%
8	1 February 2022	31 March 2022	1757.0	13.3%	1513.3	14.9%
9	1 March 2022	30 April 2022	1022.7	13.3%	1268.8	20.9%
10	1 April 2022	31 May 2022	1336.5	25.1%	1995.5	55.3%
11	1 May 2022	30 June 2022	1325.1	41.6%	1694.9	73.5%
12	1 June 2022	31 July 2022	1230.3	63.8%	2333.7	41.8%
13	1 July 2022	31 August 2022	2146.0	11.0%	2099.4	10.0%
14	1 August 2022	30 September 2022	1616.9	7.1%	2456.1	27.7%
15	1 September 2022	31 October 2022	1361.1	17.2%	1280.1	25.3%

## Data Availability

Data (except for Twitter) processed in this study will be available with reasonable request, ending 5 years following article publication.

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
