# Peer review of "Projection of COVID-19 Positive Cases Considering Hybrid Immunity: Case Study in Tokyo"

_vaccines, 2023, doi:10.3390/vaccines11030633_

Round 1

Reviewer 1 Report

Abstract: "data forecasting" is not too correct. You may simply write "forecasting".

L50: "reached 21 million" needs reference.

L52-53: Is this the best selected method? Why it is suggested? Needs justification in comparison to other approaches.

L54-58: The target for writing this research paper is not so clear. What is the scientific gap? That your study is going to cover? Your text shall have an inductive flow driving to the justification of the clear research question of your study.

The introduction should better describe the problem context, literature review and the hypothesis based on the gap analysis of the previously published research.

L105: please clarify the concept "relative risk".

Please do not use abbreviations in the titles.

L199: "training process".

Author Response

Thank you for taking the time to review the manuscript. Please see the attached file for our point-to-point response.

Reviewer 2 Report

Kodera et al use a hybrid epidemiological model to estimate hybrid immunity and vaccine efficacy in Tokyo.  Including parameters for waning immunity and differential immunity is a strength.  However there are some weaknesses, some minor some major.

Minor:

P2, line 95 toxicology is not the right word here, perhaps pathogenesis?

Figure 3:  Weakly should be weekly

Major:

They do not include seasonality in the model.  It is now well know than covid cases exhibit seasonal trends.  One could also think that mobility is also seasonal with more people meeting and congregating in the summer months than the Jan- May timeframe.

They also use baseline mobility from Feb 3- Feb 6 2020, winter months to examine changes in mobility.  Is this time frame representative of mobility in the summer months?

The parameters used should also be included in the manuscript, even as a supplemental file.

Further, their parameterization of natural immunity may need to be redone in light of the data given in Stein et al in Lancet (2023) showing that recovery from covid is at least a high, if not higher than two doses of mRNA vaccines.  Further, the waning trajectory does not seem to  be as steep. Does this change the results of the manuscript?

Is the rate of testing constant across the waves?  Is it possible that those who have already recovered may be less likely to test

Author Response

Thank you for taking the time to review our manuscript. Please see the attached file for our point-to-point response.

Reviewer 3 Report

This paper proposes a data-driven forecasting model for COVID-19 by considering hybrid immunity and makes a case study in Tokyo. The results are interesting and potentially useful. This paper is very well written with a lot of prior work done by the team. Extensive data collection was performed and a sophisticated deep learning model was applied, with a respectable accuracy of prediction. The improvement in model prediction with the consideration of hybrid immunity is reasonable.

However, a few questions and problems need to be answered and fixed.

  1. On page 3, eq. 1 has some notations, like $at$ and $s$, that need to be further explained. 
  2. On page 4, line 126, the immunity enhancement duet to hybrid immunity is assumed to be 60% or 80% of the ideal case. Are there any references or examples to support this setting?
  3. On page 4, line 135, the authors mention that the herd immunity threshold for SARS-CoV-2 was 50%-83%. Apparently, for new variants, such numbers will be unrealistic. Thus, I recommend that the author read the following literature and derive the immunity threshold in a new way. https://www.mdpi.com/1999-4915/14/7/1482
  4. On page 5, line 164, I believe '2023' should be '2022'. The same typo can be found in other places.
  5. In figure 3, there are a lot of blanks in Tokyo's data. How can the authors be sure the estimation is valid?
  6. In figure 4b, some predictions (purple solid and red dash lines) are far from the actual data. That's why the choice of the parameter is essential. The authors need to justify their options and make them more practical.
  7. The authors should provide more details regarding the implementation of deep learning models (In Section 2.3.). For instance, how is the “statistical evaluation of the contribution of several potential factors” (that the developed model is based on) is performed? How are the hyper-parameters of the deep learning models set?

Author Response

(The authors gave the same response as above.)

Round 2

Reviewer 3 Report

The authors have answered all the questions concerned. Based on the revision, the paper can be accepted.